# Self-cross Feature based Spiking Neural Networks for Efficient Few-shot Learning

**Qi Xu** [1]   **Junyang Zhu** [1]   **Dongdong Zhou** [1]   **Hao Chen** [1]   **Yang Liu** [1]   **Jiangrong Shen** † [2][3][4]   **Qiang Zhang** [1]

## Abstract

Deep neural networks (DNNs) excel in computer vision tasks, especially, few-shot learning (FSL), which is increasingly important for generalizing from limited examples. However, DNNs are computationally expensive with scalability issues in real world. Spiking Neural Networks (SNNs), with their event-driven nature and low energy consumption, are particularly efficient in processing sparse and dynamic data, though they still encounter difficulties in capturing complex spatiotemporal features and performing accurate cross-class comparisons. To further enhance the performance and efficiency of SNNs in few-shot learning, we propose a few-shot learning framework based on SNNs, which combines a self-feature extractor module and a cross-feature contrastive module to refine feature representation and reduce power consumption. We apply the combination of temporal efficient training loss and InfoNCE loss to optimize the temporal dynamics of spike trains and enhance the discriminative power. Experimental results show that the proposed FSL-SNN significantly improves the classification performance on the neuromorphic dataset N-Omniglot, and also achieves competitive performance to ANNs on static datasets such as CUB and miniImageNet with low power consumption.

## 1. Introduction

Deep neural networks (DNNs) (Krizhevsky et al., 2012; Szegedy et al., 2015; He et al., 2016) have demonstrated remarkable performance across various computer vision tasks, such as object detection and sematic segmentation. DNNs rely on large-scale labeled datasets and deep architectures, making them computationally expensive and energy-intensive, which limits their scalability. Real-world data is often scarce, dynamic, or non-stationary, which makes few-shot learning with low energy consumption increasingly important. Few-shot learning (FSL) (Fei-Fei et al., 2006; Vinyals et al., 2016) enables models to generalize from only few examples, which adapts to ever-changing conditions. Moreover, Spiking Neural Networks (SNNs), recognized as the third generation neural networks, are particularly well-suited to few-shot learning due to the exceptionally low energy consumption (Bu et al., 2023; Ding et al., 2022; Ma et al., 2024; Hu et al., 2023; Liu et al., 2024a;b; 2025b) compared to Artificial Neural Networks (ANNs) by leveraging event-driven nature. Furthermore, the spatiotemporal dynamics of SNNs allow them effectively processing sparse and temporal data, which are common in open environments. This paper investigates the potential of SNNs to enhance few-shot learning in these challenging circumstances, offering a promising solution to real-world applications that require both adaptability and efficiency.

Few-shot learning focuses on training models to address the challenge of learning with minimal labeled data. Typical FSL approaches by ANNs rely on extensive parameter optimization (Finn et al., 2017; Ravi & Larochelle, 2017; Yoon et al., 2018) or metric learning (Qiao et al., 2019; Jiang et al., 2020; Oreshkin et al., 2018), leveraging knowledge acquired during the training phase to classify new data points with few examples. However, a significant drawback of these ANN-based methods is their high computational cost and energy consumption. Since ANNs depend heavily on floating-point operations, they incur substantial energy costs, which can be prohibitive for deployment in energy-constrained facilities such as embedded systems or Internet of Things (IoT) devices (Madakam et al., 2015; Guo et al., 2023b; Liu et al., 2024b). Additionally, the reliance on large datasets and backpropagation algorithms (Cilimkovic, 2015)

[1]School of Computer Science and Technology,Dalian University of Technology,Dalian,China [2]Faculty of Electronic and Information Engineering, Xi'an Jiaotong University [3]National Key Lab of Human-Machine Hybrid Augmented Intelligence, Xi'an Jiaotong University [4]State Key Lab of Brain-Machine Intelligence, Zhejiang University. Correspondence to: Jiangrong Shen <jr-shen@zju.edu.cn>.

*Proceedings of the 42nd International Conference on Machine Learning*, Vancouver, Canada. PMLR 267, 2025. Copyright 2025 by the author(s).

for effective learning limits the generalization capability in few-shot learning scenarios.

To address these issues, researchers have turned to SNNs (Maass, 1997; Cao et al., 2015; Sorbaro et al., 2020; Guo et al., 2023a; Guan et al., 2023; Xu et al., 2023; 2024a; Shen et al., 2025; 2024), which emulate the spiking mechanism of biological neurons and offer the potential for significant reductions in computational cost and energy consumption. By encoding information through discrete spike trains and performing computations in an event-driven manner, SNNs exhibit an inherent advantage in energy efficiency(Xu et al., 2024b), making them promising candidates for resource-constrained applications. Recent studies have started to explore SNNs for few-shot learning by utilizing spike-time encoding and spike sequence processing to enable learning from limited data. However, existing SNN-based few-shot learning methods still face several challenges that need to be addressed. Firstly, SNNs often struggle to effectively capture features from complex spatiotemporal features of inputs, which limits their capacity to extract rich image features. Secondly, current approaches lack of cross-class feature comparison and classification, making it difficult to accurately map query samples to the appropriate support sets.

Given these challenges, this paper introduces a novel network to leverage SNNs for spatiotemporal feature extraction in few-shot learning tasks. Our approach integrates a self-feature extractor module and a cross-feature contrastive module to further refine feature representation and classify the query set into a specific support set. The proposed method achieves a significant advancement by combining the energy-efficient SNN with an innovative feature extraction mechanism, thereby enhancing feature representation capability and classification accuracy in few-shot learning. By addressing the high energy consumption of traditional ANN approaches and improving the performance of SNNs in few-shot scenarios, our approach achieves both outranging performance and practical significance. Our contributions are summarized as follows:

- This paper proposes a few-shot learning framework based on spiking neural networks, which combines self-feature extractor module and a cross-feature contrastive module to further refine feature representation and greatly reduce power consumption.

- By combining TET Loss and InfoNCE Loss, our model shows strong generalization ability and noise resistance. TET Loss reduces computational overhead and speeds up training, while InfoNCE Loss improves the model's ability to distinguish between positive and negative samples, enhancing robustness to noisy data.

- We fully demonstrate the model's effectiveness through experiments on static and neuromorphic datasets. Particularly, we are the first to report an accuracy of 98.9% on 5w5s on N-Omniglot and achieve great performance close to ANNs on CUB and *mini*ImageNet dataset, which sets a new state-of-the-art performance of SNN-based methods.

## 2. Related Work

Few-shot learning refers to the issue of training models to generalize from a limited number of examples. Unlike traditional machine learning tasks, where large datasets with thousands of labeled examples are available for training, FSL aims to train a model to recognize novel classes using only few labeled instances. This challenge is particularly relevant in scenarios where acquiring a large amount of labeled data is impractical or expensive, such as medical image analysis, where annotations require expert knowledge, or personalized applications, where user-specific data is scarce.

**ANN FSL** Recent studies have demonstrated various few-shot learning methods utilizing ANNs. For instance, QS-Former (Wang et al., 2023) enhances few-shot classification performance through a unified Query-Support Transformer architecture, effectively integrating global and local feature extraction. Similarly, DT-FSL (Ran et al., 2023) merges deep Transformer models with meta-learning strategies to improve hyperspectral image classification, applying domain adaptation techniques to mitigate distribution discrepancies. CTFSL (Peng et al., 2023) combines convolutional neural networks (CNNs) with Vision Transformers to achieve cross-domain hyperspectral image classification, enhancing generalization through domain alignment and discrimination. Furthermore, Luo (Luo et al., 2023) suggests that the independence of training and adaptation algorithms can simplify the development of few-shot learning techniques. A polyp classification system (Krenzer et al., 2023) based on deep learning has also shown improved diagnostic accuracy by leveraging few-shot and deep metric learning. Furthermore, EASY (Bendou et al., 2022) employs a multi-backbone network and Y-shaped structure in training, achieving advanced few-shot classification performance across multiple standard datasets. These studies collectively illustrate that ANNs could achieve good performance in few-shot learning.

**SNN FSL** Recent studies highlight the wide application and potential of SNNs in few-shot learning. Efficient online few-shot learning (Stewart et al., 2020a) has been applied to the Intel Loihi neuromorphic processor through proxy gradient descent and transfer learning for gesture recognition. Based on this, the natural e-prop method is proposed to facilitate one-shot learning with the learning signals emitted by recursive SNNs, aligning closely with biological

learning mechanisms and allowing rapid adaptation to new tasks (Scherr et al., 2020). Furthermore, the development of SOEL system (Stewart et al., 2020b), which integrates proxy gradient descent with error-triggered learning, has significantly improved the efficiency of online learning on neuromorphic hardwares. Jiang (Jiang et al., 2021) proposes a multi-timescale optimization framework, introducing an adaptive-gated LSTM to balance short-term learning with long-term evolution to acquire prior knowledge through example-level learning and task-level optimization. Stewart proposes a method that combines of meta-learning (MAML) and proxy gradient (Stewart & Neftci, 2022) to improve the rapid learning capabilities of SNN, making it particularly suitable for low-power and fast adaptation scenarios. HES-FOL (Yang et al., 2022) boosts the robustness and accuracy of SNNs in non-Gaussian noise through a heterogeneous integrated loss function. Complementing this research, a bionic SNN designed for gas recognition (Huo et al., 2023) supports the incremental learning of few-shot classes while effectively addressing sensor drift and aging issues. These studies demonstrate that SNNs could achieve both efficient performance and strong adaptability in few-shot learning.

## 3. Preliminary

### 3.1. Spiking neuron model

Leaky Integrate-and-Fire (LIF) (Shamsi et al., 2017) is the most commonly used neuron model in SNNs. It captures the dynamic characteristics of spikes by simulating the behavior of biological neurons. The LIF model mimics how neurons work and generate spikes with membrane potential and a certain threshold. The whole process can be described as:

$$\boldsymbol{U}(t) = \tau \boldsymbol{U}(t-1) + \boldsymbol{X}(t), \tag{1}$$

$$\boldsymbol{S}(t) = \Theta(\boldsymbol{U}(t) - V_{th}), \tag{2}$$

$$\boldsymbol{U}(t) = \boldsymbol{U}(t) \cdot (1 - \boldsymbol{S}(t)), \tag{3}$$

where $\tau$ is a constant leaky factor, $\boldsymbol{U}(t)$ is the membrane potential at time t, $\boldsymbol{X}(t)$ is the input, and $\Theta$ represents the Heaviside step function. Given a certain threshold $V_{th}$, if $\boldsymbol{U}(t)$ exceeds this threshold, the neuron generates a spike, then $\boldsymbol{U}(t)$ is reset to 0. The firing function and hard reset mechanism can be described as: when $\boldsymbol{U}(t) \geq V_{th}$, the neuron generates a spike, and then $\boldsymbol{U}(t)$ is reset to 0 and enters a recovery period until the membrane potential rises to the threshold again; when $\boldsymbol{U}(t)$ reaches a certain threshold $V_{th}$, the output spike $\boldsymbol{S}(t)$ is triggered by the step function. LIF neurons are often used in SNNs to replace the activation function in traditional ANNs with simulating the behavior of biological neurons.

### 3.2. Few-shot Classification

Few-shot classification aims to solve the problem of effectively classifying new samples when there are only few labeled samples in each category. Due to the extremely limited labeled data, traditional deep learning methods are prone to overfit in this case and fail to unseen data. To resolve this issue, few-shot learning usually adopts a meta-learning framework and improves the adaptability through episodic training.

In few-shot classification, FSL is usually defined as a N-way K-shot task (i.e., K labeled samples of N unique classes) and K is very small, e.g., 1 or 5. The model is initially trained on training data $\mathcal{D}_{train}$ from a set of training classes $\mathcal{C}_{train}$, and then evaluated on test data $\mathcal{D}_{test}$ from unseen classes $\mathcal{C}_{test}$, where $\mathcal{C}_{train} \cap \mathcal{C}_{test} = \varnothing$, which is the most important setting for few-shot learning. Both $\mathcal{D}_{train}$ and $\mathcal{D}_{test}$ consist of multiple episodes, each containing a query set $Q = \{(x_j, y_j)\}_{j=1}^{N \times K}$ and support set $S = \{(x_i, y_i)\}_{i=1}^{N \times K}$ of K image-label pairs for each of the N classes, also known as an N-way K-shot episode. During training, we iteratively sample an episode from $\mathcal{D}_{train}$ and train the model to learn a mapping from the support set $S$ and query images $I_q$ to the corresponding query labels $y_q$. This process involves learning how to generalize from the limited information provided by the support set to predict accurately on the query set. During testing, the model uses the learned mapping to classify query images $I_q$ as one of the $N$ classes in the support set $S$ sampled from $\mathcal{D}_{test}$. The goal is to evaluate the model's ability to generalize to new, unseen classes with limited data.

## 4. Methodology

In this section, we will introduce our network model architecture, which uses a spiking neural network to extract image features. The self-feature extractor module and the cross-feature contrastive module further extract features and classify the query set into a certain support set. The entire network model architecture is shown in the figure 1. We first briefly introduce the whole model structure, then specifically introduce the technical details of each module, and finally explain our specific training strategy.

### 4.1. Model Architecture

First, support set images **spt** and query set images **qry** are sent to the spiking network backbone to extract the basic feature representation, where the weights of the two are shared, and then enter the self-feature extractor(SFE) module to analyze the internal correlation of the image, and finally enter the cross-feature contrastive(CFC) module to classify the query set into the nearest support set. We named our model SSCF (Spiking Self-Cross Feature Network).

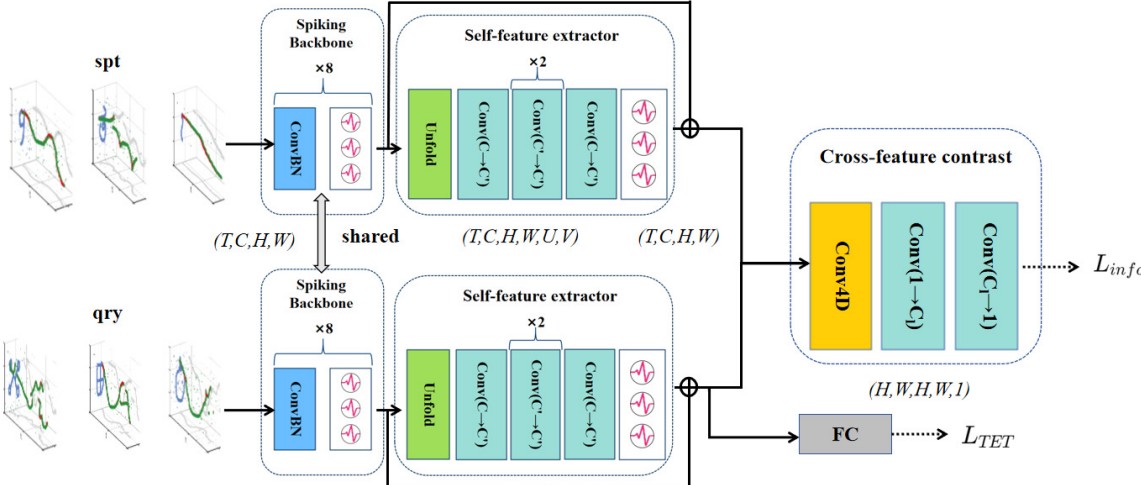

*Figure 1.* Architecture of the proposed few-shot learning framework based on SNNs. It combines self-feature extractor module and a cross-feature contrastive module to further refine feature representation and greatly reduce power consumption.

## 4.2. Backbone

The backbone network we use is VGGSNN, specifically, the spiking form of VGG16, a total of 8 conv-bn-LIF layers. Note that the backbone we use is relatively simple because an overly complex backbone is prone to overfitting, which is not conducive to the generalization ability of few-shot learning. We did not apply the average pooling layer since the average pooling operation would lose spatial information. A key feature of SNN is that they can operate sparsely, in other words, neurons will only generate spikes when the input stimulus exceeds a certain threshold. Avoiding the use of average pooling could help maintain this sparsity and may be more efficient in hardware implementation and reduce unnecessary calculations. For the input format, we replicate the static dataset $T$ time steps to meet the input requirements of our spiking backbone. For the neural morphology dataset, due to its natural $T$-dimension, there is no need to perform excessive operations. We will temporarily refer to the preliminary features obtained through the spiking backbone network as $\mathbf{F}_0$.

## 4.3. Self Feature Extractor(SFE)

The SFE module pays more attention to the information inside the image and provides a reliable input for the CFC module. Given a feature representation $\mathbf{F}_0 \in \mathbb{R}^{T \times C \times H \times W}$, where $T$ is the timestep, $C$ is the channel, and $H \times W$ denotes the spatial resolution. First, we use the unfolded operation to make each position $x \in [1, H] \times [1, W]$ in $T \times C$ expanded $U$ and $V$ dimension. Hence, it turns to $\mathbf{F}_0 \in \mathbb{R}^{T \times C \times H \times W \times U \times V}$. We use time-channel self-

correlation to retain the rich semantics of the feature vector for classification, which suppresses changes in appearance and reveals structural patterns.

Then, we apply a convolutional block that follows the computational efficiency bottleneck structure to obtain the self-correlation pattern in $\mathbf{F}_1$, namely a $1\times1$ convolution layer for reducing the number of channels and two $3\times3$ convolution layers for transformation, and a $1\times1$ convolution layer to restore the size of the number of channels. This series of convolution operations gradually aggregates local correlation patterns without padding, reducing the dimension of $U \times V$ to $1 \times 1$. After that, the LIF neuron layer is inserted as the activation function. For the entire SFE module, the feature dimension remains unchanged as $T \times C \times H \times W$. This feature extraction is complementary to the feature $\mathbf{F}$ obtained after the spiking backbone, so we combine the representation of the two modes, applying the residual structure so that the features sent to the CFC module are the sum of the two, which strengthens the representation of basic features with relational features and helps few-shot learning better understand "observe yourself for what" in the image:

$$\mathbf{F} = \mathbf{F}_0 + \mathbf{F}_1 . \tag{4}$$

This approach helps in recognizing intra-class features and facilitates generalization to unseen target categories.

## 4.4. Cross Feature Contrastive(CFC)

The CFC module takes the input pairs $\mathbf{F}_s$ and $\mathbf{F}_q$ of the support set and the query set and generates the corresponding attention maps $\mathbf{A}_q$ and $\mathbf{A}_s$. We first take the average

value of the T dimension to facilitate subsequent operations. Then, the query and support representations $\mathbf{F}_q$ and $\mathbf{F}_s$ are converted into more compact representations using point-wise convolutional layers, reducing the channel dimension $C$ to $C'$. We construct a four-dimensional cross-correlation tensor $\mathbf{C} \in \mathbb{R}^{H \times W \times H \times W}$. Since the dimensions $H$ and $W$ have been reduced after the convolutional layers of the spiking backbone, the memory usage of the cross-correlation tensor is not particularly large. However, due to some large appearance changes in the few-shot learning setting, we adopt a convolutional matching process. The tensor is further refined using 4D convolutions with matching kernels, specifically consisting of two 4D convolutional layers. The first convolution generates multiple correlation tensors with multiple matching kernels, increasing the number of channels to $C_1$, and the second convolution aggregates them into a single 4D cross-correlation tensor. From the refined cross-correlation tensor $C$, we generate joint attention maps $\mathbf{A}_q$ and $\mathbf{A}_s$, which reveal the relevant content between the query set and the support set. The attention map for a query $\mathbf{A}_q \in \mathbb{R}^{H \times W}$ is computed by (Kang et al., 2021)

$$\mathbf{A_q}(\mathbf{x_q}) = \frac{\text{softmax}(C(\mathbf{x_q}, \mathbf{x_s})/\gamma)}{HW}, \tag{5}$$

where x is the position on the feature map and $\gamma$ is the temperature factor. The attention value $\mathbf{A_q}(\mathbf{x_q})$ can be interpreted as the ratio of the matching score of $\mathbf{x_q}$, as the average probability of matching the position at the query image with the position at the support image. Similarly, the attention map for the support is computed by switching the query and support in equation 5. These joint attention maps improve the accuracy of few-shot classification by cross-correlating patterns and adjusting "important locations of joint attention" with respect to the image given at the test time.

### 4.5. Training Losses

We train the network in a single-stage way, combining two losses to guide the model to classify precisely: TET-based loss and contrast-based loss. First, we append a fully connected classification layer after F to calculate $L_{\text{TET}}$, which guides the model to correctly classify queries of class $c \in \mathcal{C}_{\text{train}}$. The contrast-based metric loss $L_{\text{info}}$ calculates the cosine similarity between the query and the support prototype embeddings, separates the positive and negative classes, and finally calculates the infoNCE loss to map the query embedding to the support embedding of the same class. When inferencing, the query class is predicted to be the class of the closest support set.

Since SNN has an additional time dimension than ANNs, our learning goal should be reconsidered. We use TET loss to train our backbone network. It turns out that TET loss is effective for spiking neural networks. The calculation of

TET loss is(Deng et al., 2022):

$$\mathcal{L}_{\text{TET}} = -\frac{1}{T} \sum_{t=1}^{T} L_{\text{CE}}(\mathbf{F_q}, \mathbf{y}), \tag{6}$$

where T is the time step, $L_{\text{CE}}$ represents the cross-entropy loss, and y is the sample label.

Next, we use the pooling to obtain the final feature expression based on the contrast metric loss. The support set is first divided into positive and negative classes according to the labels so that the network can better learn the correct category, and then the following contrast loss is calculated:

$$\mathbf{S} = \mathbf{F}_s \mathbf{A}_s, \mathbf{Q} = \mathbf{F}_q \mathbf{A}_q, \tag{7}$$

$$\mathcal{L}_{\text{infoNCE}} = -\log \frac{\exp(sim(\mathbf{S}_+, \mathbf{Q})/\tau)}{\sum \exp(sim(\mathbf{S}, \mathbf{Q})/\tau)}, \tag{8}$$

where $sim(\cdot, \cdot)$ is cosine similarity and $\tau$ is a scalar temperature factor. At inference process, the class of the query is predicted as that of the nearest prototype. The final loss function combines these two losses, where $\lambda$ is a hyper-parameter that balances the loss terms:

$$\mathcal{L}_{\text{Total}} = \lambda \mathcal{L}_{\text{TET}} + (1 - \lambda)\mathcal{L}_{\text{info}}. \tag{9}$$

## 5. Experiments

### 5.1. Datasets

We adopt the following datasets for experiments: N-Omniglot, CUB-200-2011, and *mini*ImageNet. Meanwhile, as a few-shot learning study, we must ensure that the training set and the test set are different, that is, their intersection is empty.

**N-Omniglot**(Li et al., 2022) is a neuromorphic dataset built based on the original Omniglot dataset, which consists of 1623 handwritten characters from 50 different languages. Each character has only 20 different samples.

**CUB-200-2011** is a dataset focused on the fine-grained classification of birds and is widely used in the field of few-shot learning. It has a total of 200 categories, of which 100, 50, and 50 are used for training, validation, and testing, respectively.

*mini***ImageNet**(Vinyals et al., 2016) is a derived dataset from ImageNet that comprises a total of 60,000 images, which are evenly distributed across 100 different object categories. Of these categories, 64 are designated for training, 16 for validation, and 20 are reserved for testing.

### 5.2. Experimental Results

We conducted experiments on the N-Omniglot dataset under different time steps and scenario settings for performance evaluation. Experimental results are shown in Table 1. For

| method | backbone | T | 20w1s | 20w5s | 5w1s | 5w5s |
|---|---|---|---|---|---|---|
| Siamese(Koch et al., 2015) | SCNN | 4 | 53.3±1.6 | 74.9±1.6 | 72.6±0.9 | 88.4±1.0 |
| | | 8 | 50.8±0.4 | 72.2±0.8 | 74.4±0.7 | 90.7±0.1 |
| | | 12 | 49.8±1.3 | 71.3±1.0 | 69.3±0.8 | 85.7±0.6 |
| MAML(Finn et al., 2017) | SCNN | 4 | - | - | 74.4±0.7 | 90.7±0.1 |
| | | 8 | - | - | 71.1±0.3 | 88.6±0.4 |
| | | 12 | - | - | 70.3±0.9 | 87.3±0.3 |
| SSCF(ours) | SCNN | 4 | 67.0±0.8 | 79.0±0.7 | 79.3±0.7 | 95.0±0.6 |
| | | 8 | 67.6±0.6 | 80.2±0.6 | 79.5±0.8 | 95.5±0.2 |
| | | 12 | 67.8±0.5 | 79.7±0.5 | 80.0±0.3 | 96.2±0.4 |
| SSCF(ours) | VGGSNN | 4 | 83.7±0.7 | 94.8±0.3 | 94.3±0.8 | 98.6±0.3 |
| | | 8 | 84.2±0.6 | 94.7±0.3 | 94.8±0.7 | **98.9±0.3** |
| | | 12 | **84.4±0.6** | **94.9±0.2** | **95.3±0.6** | 98.8±0.2 |

*Table 1.* Performance comparison among different SNNs-based FSL models on the N-Omniglot dataset

the convenience of comparison, we conducted experiments on both the SCNN backbone network and the VGGSNN backbone network. The experimental results on the SCNN backbone network surpassed maml and siamese, proving the effectiveness of our model architecture. The experimental results on the VGGSNN backbone network were further improved, which proved the power of VGGSNN as a spiking neural backbone network.We can observe that when the number of ways remains constant, increasing the number of shots leads to a higher classification accuracy. When the number of shots remains constant, reducing the number of ways results in higher accuracy. In the 1-shot scenario, as the time step $T$ increases, the accuracy consistently improves. However, with an increase in the number of shots, such as in our 5-shot experiments, the impact of the time step $T$ on accuracy becomes minimal. This is because, in the 1-shot scenario, the limited number of samples means that a longer time step can provide more information to the model, thereby enhancing its performance. In the 5-way 1-shot (5w1s) setting, our model achieves an accuracy of 95.3% at T=12, representing a 25.0% improvement over MAML. In the 5-way 5-shot (5w5s) setting, the performance of our model is relatively stable across different time steps, with the accuracy consistently around 98.8%, which is a 11.5% improvement over MAML. In the 20-way 1-shot (20w1s) and 20-way 5-shot (20w5s) settings, our model achieves accuracies of 84.4% and 94.2%, respectively, representing improvements of 34.6% and 23.6% over the Siamese method reported in the original paper. Results show particularly significant improvements in the 1-shot setting, which further proves its strong generalization performance and suitability for solving few-shot learning problems. These results clearly demonstrate that our model performs exceptionally well on neuromorphic datasets. In figure 2, we visualize the t-SNE graph at different time steps and different ways.

Next, we evaluate the performance on static datasets.

We perform experiments on the CUB and *mini*ImageNet datasets. Results are shown in Table 2 and Table 3.On the CUB dataset, our model achieves an accuracy of 76.27% in the 5-way 1-shot (5w1s) setting and 87.00% in the 5-way 5-shot (5w5s) setting. On the *mini*ImageNet dataset, our model achieves an accuracy of 60.97% in the 5w1s setting and 75.61% in the 5w5s setting. These results surpass many classic ANNs methods and are close to the current state-of-the-art performance of ANNs.

We found that, since the CUB dataset is a fine-grained image dataset, balancing the loss terms favors self-correlation features. Given that all images in the dataset are of birds, inter-class features are less important. This does not mean that inter-class feature comparison can be omitted; both types of features still need to be appropriately balanced. The *mini*ImageNet dataset, on the other hand, contains many classes with significant differences, making cross-class features more important than in the CUB dataset. However, the optimal $\lambda$ coefficient is still greater than 0.5, as inter-class features are derived from self-class features. Table 4 shows our experimental results with different values of $\lambda$, which supports the above observations.

### 5.3. Ablation Studies

Table 5 presents the results of our ablation study, which is conducted on the N-Omniglot and CUB datasets. Results show that both modules are beneficial to the model, with the SFE (Self-Feature Extraction) module being particularly crucial for improving the performance. Specifically, on the N-Omniglot dataset, removing the SFE module leads to a significant drop in accuracy, from 94.34% to 93.17% in the 5-way 1-shot (5w1s) setting. Similarly, on the CUB dataset, the removal of the SFE module causes a substantial decrease in accuracy, from 76.27% to 71.84% in the 5w1s setting. This indicates that the SFE module is essential for capturing

| method | backbone | 5-way 1-shot | 5-way 5-shot |
|---|---|---|---|
| ProtoNet(Snell et al., 2017) | ResNet12 | 66.09±0.92 | 82.50±0.58 |
| RelationNet(Sung et al., 2018) | ResNet34 | 66.20±0.99 | 82.30±0.58 |
| DEML+Meta-SGD(Zhou et al., 2018) | ResNet50 | 66.95±1.06 | 77.11±0.78 |
| MAML(Finn et al., 2017) | ResNet34 | 67.28±1.08 | 83.47±0.59 |
| MergeNet-MAX(Atanbori & Rose, 2022) | MobileNetV2 | 72.90±0.24 | 81.76±0.25 |
| S2M2(Mangla et al., 2020) | ResNet34 | 72.92±0.83 | 86.55±0.51 |
| FEAT(Ye et al., 2020) | ResNet12 | 73.27±0.22 | 85.77±0.14 |
| MergeNet-MAX(Atanbori & Rose, 2022) | EfficientNetB0 | 75.34±0.21 | 83.42±0.29 |
| DeepEMD(Zhang et al., 2020) | ResNet12 | 75.65±0.83 | 88.69±0.50 |
| RENet(Kang et al., 2021) | ResNet12 | **79.49±0.44** | **91.11±0.24** |
| SSCF(ours) | VGGSNN | 76.27±0.46 | 87.00±0.30 |

*Table 2.* Performance comparison on CUB dataset

| method | backbone | 5-way 1-shot | 5-way 5-shot |
|---|---|---|---|
| MAML(Finn et al., 2017) | ResNet34 | 48.70±1.84 | 63.11±0.92 |
| Meta-SGD(Zhou et al., 2018) | ResNet50 | 50.47±1.87 | 64.66±0.89 |
| adaResNet(Munkhdalai et al., 2018) | ResNet12 | 56.88±0.62 | 71.94±0.57 |
| RelationNet(Sung et al., 2018) | ResNet34 | 57.02±0.92 | 71.07±0.69 |
| Dual TriNet(Chen et al., 2019) | ResNet18 | 58.12±1.37 | 76.92±0.69 |
| ProtoNet(Snell et al., 2017) | ResNet12 | 62.39±0.21 | 80.53±0.14 |
| S2M2(Mangla et al., 2020) | ResNet34 | 63.74±0.18 | 79.45±0.12 |
| DeepEMD(Zhang et al., 2020) | ResNet12 | 65.91±0.82 | 82.41±0.56 |
| RENet(Kang et al., 2021) | ResNet12 | **67.60±0.44** | **82.58±0.30** |
| SSCF(ours) | VGGSNN | 60.97±0.45 | 75.61±0.34 |

*Table 3.* Performance comparison on *mini*ImageNet dataset

| $\lambda$ | 0.2 | 0.4 | 0.6 | 0.7 | 0.8 | 1.0 |
|---|---|---|---|---|---|---|
| cub | 45.98 | 66.67 | 73.56 | **76.27** | 72.17 | 68.37 |
| *mini* | 35.89 | 50.02 | **60.97** | 60.71 | 60.74 | 53.19 |

*Table 4.* Performance comparison under different $\lambda$ on CUB and *mini*ImageNet with time steps T=2

| noise | 0.0 | 0.4 | 0.8 |
|---|---|---|---|
| **CE**(Wang et al., 2019) | 55.135 | 40.649 | 32.725 |
| **infoNCE**(He et al., 2020) | 55.302 | 43.699 | 36.485 |

*Table 6.* Performance under different levels of noise on CUB

| SFE | CFC | N-Omniglot | CUB |
|---|---|---|---|
| ✓ | ✓ | 92.13 | 69.97 |
| × | ✓ | 94.07(+1.94) | 72.20(+2.23) |
| ✓ | × | 93.17(+1.04) | 71.84(+1.87) |
| ✓ | ✓ | **94.34(+2.21)** | **76.27(+6.30)** |

*Table 5.* Ablation studies on SFE and CFC components

fine-grained details and improving classification accuracy, especially in datasets with closely related classes. While the other module also contributes to the model's performance, its impact is less pronounced compared to the SFE module. These findings highlight the importance of both modules and demonstrate the effectiveness of our model in handling both fine-grained and general classification tasks.

The primary challenge in few-shot learning is the lack of rich and diverse data. For instance, in identifying rare animal species and rare diseases, the environments in which we collect data are often complex and difficult, leading to in-

evitable noises. To address this issue, we modified the data preprocessing steps and introduced Gaussian noise to the dataset. Specifically, we selected the bird dataset CUB and added noise rates of 0.0, 0.4, and 0.8 for our experiments. The comparison results, shown in Table 6, demonstrate that using the infoNCE loss function yields better performance compared to using only the ordinary cross-entropy (CE) loss. At a noise rate of 0.0, the model using infoNCE achieves an accuracy of 55.302%, while the model using CE achieved 55.135%. As the noise rate increases to 0.4, the infoNCE model maintains a higher accuracy of 43.699%, whereas the CE model's accuracy drops to 40.649%. Even at the highest noise rate of 0.8, the infoNCE model outperforms the CE model with an accuracy of 36.485% compared to 32.725%. These results clearly indicate that our model, when using the infoNCE loss, exhibits superior robustness and is better equipped to handle noisy data, which is a common issue in few-shot learning scenarios.

Our model has significant advantages over traditional ANN

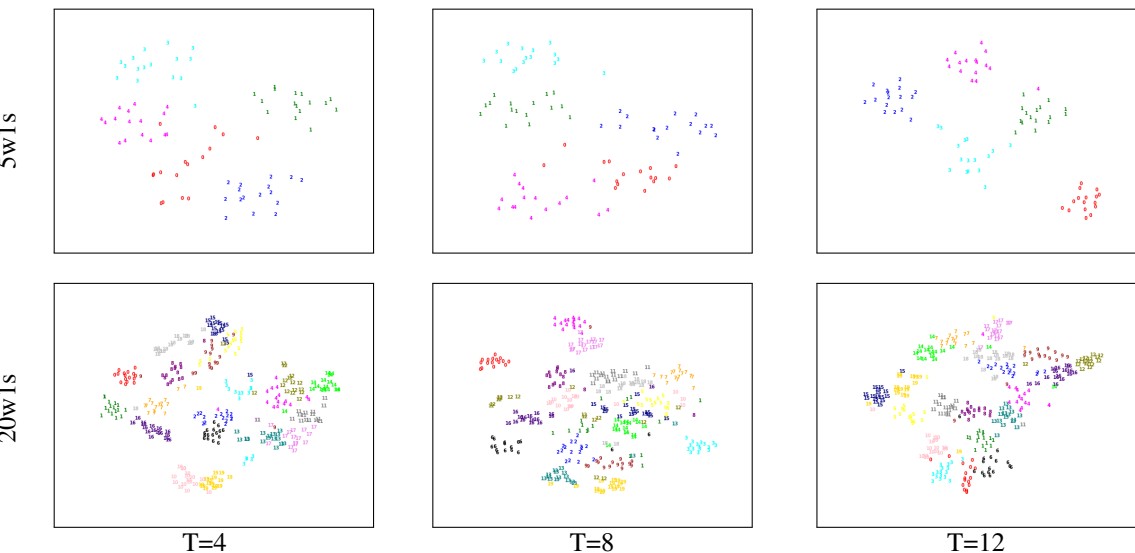

*Figure 2.* T-SNE visualization on N-Omniglot in different time steps.We can intuitively see the clustering effect in the feature space and the impact of the time step on the feature distribution. At a shorter time step (such as T=4), the feature distribution is more dispersed and the distinction between categories is low. As the time step increases (such as T=12), the feature distribution gradually becomes more compact and the distinction between categories is significantly improved.

| Method | acc | SOP | FLOPs | Energy |
|--------|------|-------|-------|----------|
| ReNet | 79.49 | - | 4.84G | 22.264mJ |
| **ours** | 76.27 | 1.39G | 0.13G | 1.849mJ |

*Table 7.* Comparison of the energy consumption with ANNs

in terms of deploying on neuromorphic hardware. Next, we will perform theoretical synaptic operation and energy consumption calculation (Liu et al., 2025a). For SNNs, we must first calculate the synaptic operation and then calculate the energy consumption. The energy consumption is proportional to the number of operations. The calculation is:

$$\text{SOPs} = fr \times T \times \text{FLOPs}, \tag{10}$$

$$\text{E}_{\text{SNN}} = 0.9pJ \times \text{SOPs}, \tag{11}$$

$$\text{E}_{\text{ANN}} = 4.6pJ \times \text{FLOPs}, \tag{12}$$

where $fr$ is the firing rate of the input spike train of the block/layer and T is the simulation time step of spike neuron. 0.9pJ is the energy consumption of each SOP(Hu et al., 2021; Indiveri et al., 2015). For ANNs, 4.6pJ is the energy consumption of each FLOP. From Table 7, we can see that our experiment is compared with the ANNs method. When T=2, the energy consumption of our method is only 8.30% of the ANNs'. This proves that SNN is particularly good in power consumption optimization with its unique event-driven computing method and sparse data processing capabilities.

## 6. Conclusion

This paper mainly focuses on implementing few-shot learning with SNN to alleviate the issue of insufficient generalization ability and improve the energy efficiency. We combine self-feature extractor module and a cross-feature contrastive module to further refine feature representation and greatly reduce power consumption. Our training strategy combines TET loss and infoNCE loss at the same time. This method has good performance on the neuromorphic dataset N-Omniglot and also obtains competitive performance with ANNs on static datasets such as CUB and *mini*ImageNet. The energy estimation also shows the effiency priority of our model.

Thus, advancing FSL research not only addresses critical limitations in current AI methodologies but also meets the growing demand for adaptive, efficient, and scalable solutions in real-world applications. Future work will focus on further refining the model to improve its performance on even more challenging datasets and tasks. We aim to explore advanced techniques for feature extraction and loss function optimization to enhance the model's generalization ability. Additionally, we plan to investigate the integration of additional spiking attention-based mechanisms and temporal dynamics to better capture the temporal aspects of data, which are often crucial in real-world applications.

## Acknowledgements

This work was supported by the National Natural Science Foundation of China under Grant (No. 62306274, 62476035, 62206037, 61925603, U24B20140), and in part by the Young Elite Scientists Sponsorship Program by CAST under Grant 2024QNRC001. Open Research Program of the National Key Laboratory of Brain-Machine Intelligence, Zhejiang University (No. BMI2400012).

## Impact Statement

This paper presents work whose goal is to advance the field of Deep Learning. There are many potential societal consequences of our work, none which we feel must be specifically highlighted here.

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

## A. Supplementary Materials

In order to gain a deeper understanding of the encoder's performance during feature extraction, we extracted the encoder part of the model and added a spike visualization module to intuitively display the spike activity in the spiking neural network (SNN). With this method, we can clearly observe the activation patterns and their changing patterns of spiking neurons at different time steps. The experimental results show that within the initial time step, the spike activity is mainly focused on capturing the key features of the input data, showing the encoder's high sensitivity to important information. As the time step increases, more and more features are activated and emit spikes, indicating that the encoder can gradually identify more subtle features. This phenomenon reveals that the encoder not only maintains sensitivity to the main features at long time steps, but also gradually captures more complex detail features, thereby enhancing the overall performance of the model. The introduction of spike visualization not only helps us better understand the internal working mechanism of the encoder, but also provides valuable insights for further optimizing the model.

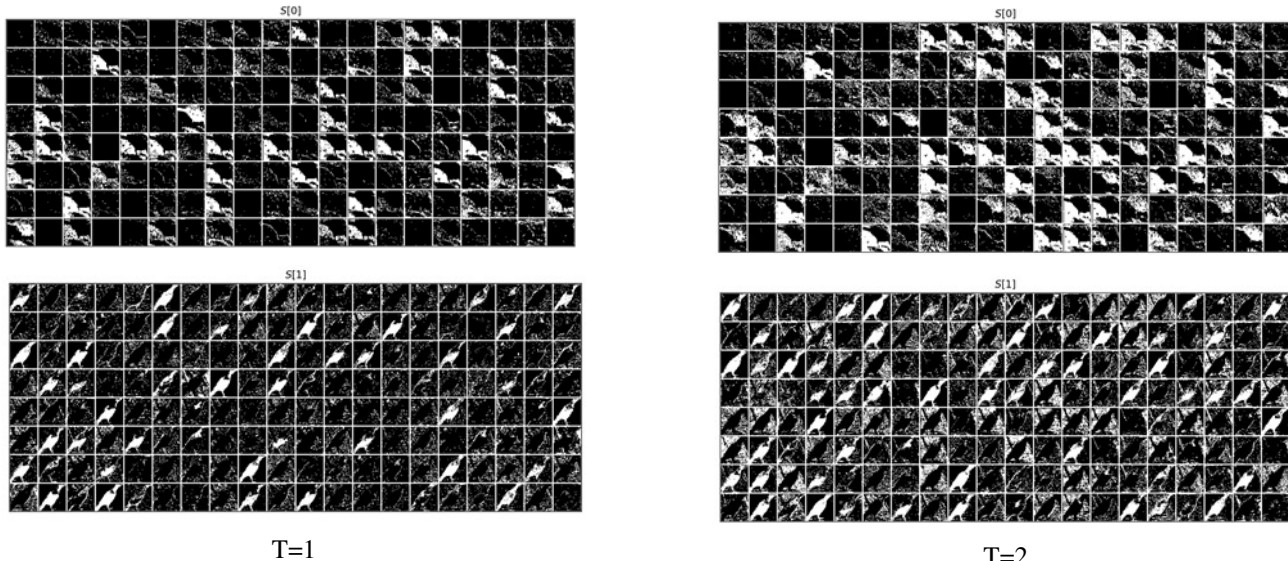

T=1

T=2

*Figure 3.* Visualization of spiking activities at different time steps.In the above picture, we can clearly see the changes in the activity of the spiking. The features at time T=2 are significantly more than those at time T=1. This shows that as the time step increases, more and more features are activated and spikings are emitted, thereby capturing more features.

In order to further explore the performance of spiking neural networks (SNNs) on static datasets, we conducted experiments with different time steps on two commonly used static datasets, CUB and miniImageNet, and used pulse visualization techniques to show the changes in pulse activity. The experimental results show that within the initial time step, the encoder mainly captures the key features of the input data, showing a high sensitivity to important information. As the time step increases, more and more features are activated and emit pulses, indicating that the encoder can gradually identify more subtle features. Specifically, on the CUB dataset, as the time step increases from 2 to 4 and 8, the model's ability to capture complex features is significantly improved, and the classification accuracy also increases accordingly. Similarly, on the miniImageNet dataset, a longer time step enables the model to better process detailed features in the image, thereby improving the overall classification performance.

| Dataset | T | acc |
|---|---|---|
| CUB | 2 | 71.43±0.48 |
| | 4 | 73.27±0.47 |
| | 8 | 76.42±0.46 |
| *mini*ImageNet | 2 | 60.26±0.45 |
| | 4 | 60.43±0.46 |
| | 8 | 60.75±0.57 |

*Table 8.* Accuracy under different $T$ values on CUB and *mini*ImageNet datasets.

In this study, we further explored the impact of different levels of Gaussian noise on model performance, especially comparing the performance of the InfoNCE loss function with the cross entropy (CE) loss function on neuromorphic datasets and static datasets. Experimental results show that under various noise conditions, whether it is a neuromorphic dataset or a static dataset, the InfoNCE loss function exhibits superior noise resistance compared to the traditional cross entropy loss function. As the noise level increases, the InfoNCE loss function not only maintains a high classification accuracy, but its advantage over the CE loss function also becomes more obvious. These results show that the InfoNCE loss function is significantly more robust than traditional methods in dealing with high noise environments, and exhibits stronger noise resistance when the noise level increases, with higher reliability and applicability.

| Dataset | Noise | Loss | Acc |
|---|---|---|---|
| N-Omniglot | 0.0 | CE | 93.8 |
| | | InfoNCE | 94.3 |
| | 0.4 | CE | 84.2 |
| | | InfoNCE | 86.5 |
| | 0.8 | CE | 75.6 |
| | | InfoNCE | 79.4 |
| CUB | 0.0 | CE | 74.1 |
| | | InfoNCE | 77.4 |
| | 0.4 | CE | 59.8 |
| | | InfoNCE | 65.7 |
| | 0.8 | CE | 36.5 |
| | | InfoNCE | 43.6 |

*Table 9.* Performance of InfoNCE and cross entropy loss functions on different datasets at different noise levels

We have also added some of the latest SNN few-shot learning methods for comparison, The performance comparison is show as follows:

| Dataset | method | backbone | Task | acc |
|---|---|---|---|---|
| N-Omniglot | plain | SCNN | | 63.4 |
| | Knowledge-Transfer(He et al., 2024) | SCNN | 20-way 1-shot | 64.1 |
| | SSCF(ours) | SCNN | | 67.8 |
| | SSCF(ours) | VGGSNN | | 84.4 |

*Table 10.* Performance comparison of recent SNN methods on N-Omniglot dataset under 20-way 1-shot classification task

| Dataset | method | backbone | Task | acc |
|---|---|---|---|---|
| *mini*ImageNet | OWOML (Rosenfeld et al., 2021) | SNN-ResNet-12 | | 45.2 |
| | CESM (Zhan et al., 2024) | SNN-WideResNet-28-10 | 5-way 1-shot | 51.8 |
| | MESM(Zhan et al., 2024) | SNN-WideResNet-28-10 | | 53.6 |
| | SSCF(ours) | VGGSNN | | 60.9 |

*Table 11.* Performance comparison of recent SNN methods on *mini*ImageNet dataset for 5-way 1-shot classification

