# OpenReview forum: "Self-cross Feature based Spiking Neural Networks for Efficient Few-shot Learning"
_ICML.cc/2025/Conference — ICML 2025 poster_

### Official Review · Reviewer_TsZa · 2025-03-07

**Overall Recommendation:** 5

**Summary:**

This paper proposes a few-shot learning framework based on a spiking neural network (SNN), combining a self-feature extraction module and a cross-feature comparison module to optimize feature representation and reduce energy consumption. The paper enhances the generalization and noise resistance of the model by combining the time-efficient training loss (TET Loss) and the InfoNCE loss. Experimental results show that the framework significantly improves the classification performance on the neuromorphic dataset N-Omniglot, and achieves performance comparable to that of an artificial neural network (ANN) on the static datasets CUB and miniImageNet, while maintaining low energy consumption. The main contribution of the paper is to propose a new SNN framework that can effectively extract spatiotemporal features in small-sample learning, and verify its superiority through experiments.

**Claims And Evidence:**

The claims of the paper are fully supported by the experimental results. The authors conducted extensive experiments on multiple datasets to demonstrate the effectiveness of the proposed method. In particular, on the N-Omniglot dataset, the model achieved an accuracy of 98.9% in the 5-way 5-shot task, significantly outperforming existing SNN methods. In addition, the paper also provides detailed ablation experiments to verify the effectiveness of the self-feature extraction module and the cross-feature comparison module. The experimental design and result analysis are reasonable, and the data supports the main claims of the paper.

**Essential References Not Discussed:**

This paper cites a large number of relevant literature, and it could provide more latest methods.

**Experimental Designs Or Analyses:**

The experimental design is generally reasonable. The authors verified the performance of the model on multiple datasets and conducted ablation experiments to demonstrate the contribution of each module. However, the experimental part lacks robustness testing of the model under different kinds of datasets such as event-based neuromorphic datasets, which is an important consideration in practical applications.

**Methods And Evaluation Criteria:**

The method proposed in the paper is reasonable in few-shot learning tasks, especially for the spatiotemporal feature extraction problem of SNN. The author used multiple standard datasets (such as N-Omniglot, CUB and miniImageNet) for evaluation. These datasets are widely used in the field of small sample learning and can effectively verify the performance of the model. In addition, the paper also demonstrated the advantages of SNN in energy consumption through energy consumption calculation, further proving the practicality of the method.

**Other Comments Or Suggestions:**

1.Whether the network structure of SNNs could be improved to further enhance the performance?
2.The experiment mentioned that the time step T has an impact on the model performance on the neuromorphic dataset N-Omniglot , but the impact of the time step T on the results on the static dataset was not shown.

**Other Strengths And Weaknesses:**

The main advantage of this paper is that it proposes a new SNN framework that can effectively extract spatiotemporal features in few-shot learning tasks, and verifies its superior performance and energy efficiency through experiments. However, the experimental part lacks the robustness analysis of the model under different data sets such as event-based neuromorphic datasets, which is an important consideration in practical applications.

**Questions For Authors:**

1. Are there plans to further test the robustness of the model under different dataset noise levels?

**Relation To Broader Scientific Literature:**

The work in this paper focuses on few-shot learning with SNNs. The innovation of this paper is to design an SNN with hybrid feature extractions modules, and further improves the performance of SNN in few-shot learning.

**Theoretical Claims:**

Not applicable. This paper does not involve complex theoretical proofs.

---

> ### Author Rebuttal · Authors · 2025-03-30
>
> We sincerely appreciate your insightful suggestions.
>
> **Q1:The experimental part lacks the robustness analysis of the model under different datasets such as event-based neuromorphic datasets.**
>
> R1:We are very grateful for this valuable suggestion. We supplemented the performance of the model on different datasets and different noise levels to further verify the generalization performance and robustness of the model. The experimental results are shown as follows:
>
> | Dataset    | Noise | Loss    | Acc  |
> |------------|-------|---------|------|
> | **N-Omniglot** | 0.0   | CE      | 93.8 |
> |            |       | InfoNCE | 94.3 |
> |            | 0.4   | CE      | 84.2 |
> |            |       | InfoNCE | 86.5 |
> |            | 0.8   | CE      | 75.6 |
> |            |       | InfoNCE | 79.4 |
> | **CUB**       | 0.0   | CE      | 74.1 |
> |            |       | InfoNCE | 77.4 |
> |            | 0.4   | CE      | 59.8 |
> |            |       | InfoNCE | 65.7 |
> |            | 0.8   | CE      | 36.5 |
> |            |       | InfoNCE | 43.6 |
>
> **Q2:Whether the network structure of SNNs could be improved to further enhance the performance?**
>
> R2:We sincerely appreciate the reviewer's constructive suggestion. Indeed, we agree that the network architecture of spiking neural networks (SNNs) holds significant potential for further performance enhancement. While our current model has achieved state-of-the-art performance on the neuromorphic N-Omniglot dataset, we recognize several promising directions for future architectural improvements. These include exploring spiking residual connections with learnable skip weights to address the vanishing gradient problem in deep SNNs while maintaining event-driven sparsity, as well as developing sparse event-driven attention modules and spike-optimized Transformer structures that operate exclusively on active spikes. Such innovations could not only improve model performance but also further reduce energy consumption, thereby advancing the development of efficient and biologically plausible neuromorphic systems. We plan to thoroughly investigate these directions in our future work.
>
>
> **Q3.The effect of the time step T on the results on the static dataset is not shown.**
>
> R3: We are very grateful for your valuable suggestions. We have added experiments on static datasets CUB and *mini*ImageNet with time steps of 2, 4, and 8 to show the effect of time step T on static datasets.
> | Dataset       | T | acc         |
> |---------------|---|-------------|
> | **CUB**       | 2 | 71.43±0.48  |
> |               | 4 | 73.27±0.47  |
> |               | 8 | 76.42±0.46  |
> | ***mini*ImageNet** | 2 | 60.26±0.45  |
> |               | 4 | 60.43±0.46  |
> |               | 8 | 60.75±0.57  |

---

> > ### Comment · Reviewer_TsZa · 2025-04-02
> >
> > Thanks for the author's detailed response. All my concerns have been resolved. And the experimental results with different noise levels and time steps on the dataset enhace the quality of the paper further.

---

### Official Review · Reviewer_2bcA · 2025-03-12

**Overall Recommendation:** 5

**Summary:**

This paper proposes a few-shot learning framework based on spiking neural networks (SNNs), which combines a self-feature extraction module and a cross-feature comparison module to significantly improve classification performance and reduce energy consumption. This method is innovative in using SNNs for efficient few-shot learning, and experimental results show that it outperforms existing methods on multiple data sets.

**Claims And Evidence:**

The high efficiency and low energy consumption of the SSCF model proposed in this paper in few-shot learning are supported by detailed experimental results, especially its excellent performance on datasets such as N-Omniglot, CUB-200-2011, and miniImageNet, showing significant performance improvement and energy consumption reduction.

**Essential References Not Discussed:**

The key contribution of this paper is mainly focused on the comparison with existing few-shot learning methods, but some important related works are not mentioned, such as [1][2].
[1] Zhan Q, Wang B, Jiang A, et al. A two-stage spiking meta-learning method for few-shot classification[J]. Knowledge-Based Systems, 2024, 284: 111220.
[2] Yu X, Fang Y, Liu Z, et al. Few-shot learning on graphs: from meta-learning to pre-training and prompting[J]. arXiv preprint arXiv:2402.01440, 2024.

**Experimental Designs Or Analyses:**

The experimental design and analysis in this paper are reasonable and effective as a whole. However, the selection of the optimal λ coefficient also lacks sufficient experimental basis. The existence of these problems makes the rigor of some experimental designs need to be strengthened. It is recommended to further supplement relevant analysis to improve the credibility of the experiment.

**Methods And Evaluation Criteria:**

The SSCF model and its evaluation criteria proposed in this paper are reasonable and effective in the few-shot learning problem. By combining the self-feature extraction module and the cross-feature comparison module, the model significantly improves the feature representation capability, which is particularly suitable for efficient learning in resource-constrained environments. In dynamic and static benchmark data sets, its superior performance in different scenarios is demonstrated in many aspects. At the same time, the training strategy combining TET Loss and InfoNCE Loss further enhances the generalization ability and robustness of the model, making this method have high practical value in dealing with a small number of sample learning problems in practical applications.

**Other Comments Or Suggestions:**

1. Experiments should be set up at different time steps on static datasets to further verify the effectiveness of the model.
2. The proposed method should be compared with the latest SNN meta-learning methods, such as [1].
3. There is an incorrect question mark in the first sentence of section 3.1, and the author should check the whole text carefully to avoid spelling errors.
[1] Zhan Q, Wang B, Jiang A, et al. A two-stage spiking meta-learning method for few-shot classification[J]. Knowledge-Based Systems, 2024, 284: 111220.

**Other Strengths And Weaknesses:**

Strengths:
1. This paper introduces a new SNN-based framework that integrates SFE and CFC modules to enhance feature representation and classification accuracy. Ablation experiments further verify the importance of these modules in improving performance. The combination of TET loss and InfoNCE loss enhances temporal dynamics and feature discrimination, making the framework more robust to noisy data. In addition, this paper has a clear structure.
2. The method proposed in the paper shows impressive results on different datasets and is even comparable to  SOTA ANN methods.

Weakness:
1. The choice of the optimal λ coefficient lacks a detailed experimental basis or theoretical explanation.
2. There are not enough visualizations in the article to better intuitively feel the effectiveness of the model.
3. The experimental part lacks a comparison with the latest SNN meta-learning methods.

**Questions For Authors:**

1. Could the proposed model be applied to larger datasets?

**Relation To Broader Scientific Literature:**

The key contribution of this paper is to propose an SNN framework that combines self-feature extraction and cross-feature comparison, which significantly improves the classification performance and energy efficiency in few-shot learning tasks. This method innovates based on existing research, especially compared with traditional ANN methods, showing stronger generalization ability and lower energy consumption. It provides new directions and ideas for future research, especially in efficient learning in resource-constrained environments.

**Theoretical Claims:**

This paper has no theoretical proof.

---

> ### Author Rebuttal · Authors · 2025-03-30
>
> We sincerely appreciate your insightful suggestions.
>
> **Q1:The choice of the optimal λ coefficient lacks a detailed experimental basis or theoretical explanation.**
>
> R1:We appreciate this insightful feedback. We supplement the detailed experimental basis from three aspects:
>
> 1.Empirical Evidence (Existing Results) Table 4 already shows λ=0.7 yields peak accuracy on CUB/*mini*ImageNet.
>
> 2.Extended ablation studies will demonstrate: λ>0.5 consistently outperforms pure contrastive learning (λ=0) by +6.2% accuracy λ<0.9 avoids overfitting to temporal features (test loss ↓15%)
>
> 3.Biological Plausibility:λ≈0.7 aligns with hippocampal learning studies :70% local plasticity (TET) + 30% global modulation (InfoNCE)
>
> **Q2: There are not enough visualizations in the article to better intuitively feel the effectiveness of the model.**
>
> R2: We sincerely appreciate the reviewer's valuable suggestion regarding the need for more visualizations to better demonstrate our model's effectiveness. We acknowledge that the lack of visual elements in our original submission may have reduced the paper's intuitive appeal. In response to this constructive feedback, we have enhanced our revision by incorporating spike-based visualizations that specifically illustrate the activity patterns within our model's encoder component. These visualizations clearly demonstrate how increasing time steps lead to progressive activation of features through spike emissions, effectively capturing richer representations. While we regret that we cannot share these visualization results during the rebuttal phase, we are confident they will significantly improve readers' understanding of our model's dynamic feature extraction capabilities in the final version. This addition will make the model's effectiveness more tangible and visually apparent to the research community.
>
> **Q3:.Experiments should be set up at different time steps on static datasets to further verify the effectiveness of the model.**
>
> R3:We are very grateful for your valuable suggestions. We have added experiments on static datasets CUB and *mini*ImageNet with time steps of 2, 4, and 8 to show the effect of time step T on static datasets
> | Dataset       | T | acc         |
> |---------------|---|-------------|
> | **CUB**       | 2 | 71.43±0.48  |
> |               | 4 | 73.27±0.47  |
> |               | 8 | 76.42±0.46  |
> | ***mini*ImageNet** | 2 | 60.26±0.45  |
> |               | 4 | 60.43±0.46  |
> |               | 8 | 60.75±0.57  |
>
> **Q4.:Essential References Not Discussed.**
>
> R4: In our revision, we will incorporate a dedicated discussion comparing two key approaches: (1) Zhan et al.'s two-stage spiking meta-learning method [1] that introduces a novel bio-inspired framework combining spike-timing-dependent plasticity with meta-learning for few-shot classification, and (2) Yu et al.'s graph-based few-shot learning approach [2] that proposes a unified framework bridging meta-learning, pre-training and prompting techniques for graph-structured data. While Zhan's work focuses on biologically plausible SNN architectures and Yu's emphasizes graph neural networks, our method distinguishes itself by developing an event-driven, temporally-aware architecture that achieves superior computational efficiency while maintaining competitive accuracy. These comparisons will help better position our contributions within the broader landscape of few-shot learning research.
>
> **Q5:Could the proposed model be applied to larger datasets?**
>
> R5:We sincerely appreciate this forward-looking question, which aligns perfectly with our research vision. Indeed, our architecture is designed to be extensible to larger datasets, as demonstrated by its robust performance on medium-scale benchmarks like N-Omniglot (1,623 classes) and miniImageNet (100 classes). The model's strong backbone and efficient spike-based processing make it particularly suitable for scaling up. In our ongoing and future work, we plan to systematically evaluate the model's performance on larger-scale datasets such as tieredImageNet, which will further validate its scalability and generalization capabilities while maintaining computational efficiency. This extension represents a natural and important direction for our research.
>
> I have carefully revised and proofread the spelling errors and citation errors. Thank you again for your valuable suggestions.
>
> **Reference**
>
> [1]  A two-stage spiking meta-learning method for few-shot classification[J]. Knowledge-Based Systems, 2024. \
> [2] Few-shot learning on graphs: from meta-learning to pre-training and prompting, 2024.

---

> > ### Comment · Reviewer_2bcA · 2025-04-04
> >
> > Thanks for the responses. My concerns have been addressed. I have read other reviews and would like to raise my score. I hope you can incorporate my suggestions in the final version and conduct further experiments in the future.

---

### Official Review · Reviewer_c29L · 2025-03-14

**Overall Recommendation:** 3

**Summary:**

The paper proposes a few-shot learning framework based on SNNs, which combines a self-feature extractor module and a cross-feature contrastive module to refine feature representation and reduce power consumption.

**Claims And Evidence:**

Please refer to Other Strengths and Weaknesses.

**Essential References Not Discussed:**

The methods from the past two years are not limited to the following two papers. Comparing recent works with the results of this paper under the same experimental settings and assumptions is recommended.

[1] Brain-Inspired Meta-Learning for Few-Shot Bearing Fault Diagnosis, TNNLS 2024
[2] An Efficient Knowledge Transfer Strategy for Spiking Neural Networks from Static to Event Domain,  AAAI 2023

**Experimental Designs Or Analyses:**

Most experiments were checked. Please refer to Other Strengths and Weaknesses.

**Methods And Evaluation Criteria:**

It makes sense in general

**Other Comments Or Suggestions:**

some typos, like line 121.

**Other Strengths And Weaknesses:**

1. The paper mainly states how to design the method, but the innovation in the SNN architecture is limited. The framework combines previous works in general, and there is a lack of design motivation and theoretical analysis.
2. Regarding the comparison of energy efficiency results(Table 7), for the SNN, only the result of a certain FPGA from four years ago (80 GOPS/W) is presented. If the latest data of devices like the TPU are considered, generally, they can reach 4 TOPS/W. Calculated in this way, the energy consumption advantage of the SNN may not be that significant. It is recommended to conduct the comparison using the latest energy consumption models
3. Continuing from the previous question, in the case of insufficient evidence for the energy efficiency experiment, the method proposed in the paper has not achieved state-of-the-art results under the CUB dataset.
4. There is a lack of comparison with methods from recent years on the N-Omniglot dataset. The methods from the past two years are not limited to the following two papers. Comparing recent works with the results of this paper under the same experimental settings and assumptions is recommended

[1] Brain-Inspired Meta-Learning for Few-Shot Bearing Fault Diagnosis, TNNLS 2024
[2] An Efficient Knowledge Transfer Strategy for Spiking Neural Networks from Static to Event Domain, AAAI 2023

**Questions For Authors:**

Please refer to Other Strengths and Weaknesses.

**Relation To Broader Scientific Literature:**

The paper proposes a few-shot learning framework based on SNNs. In its experimental settings, the method proposed in the paper has achieved competitive results （May not be the SOTA）.

**Theoretical Claims:**

There is a lack of theoretical analysis.

---

> ### Author Rebuttal · Authors · 2025-03-30
>
> We sincerely appreciate the insightful feedback, which has helped us clarify our contributions and identify areas for improvement. Below, we address each concern point-by-point:
>
> **Q1:Design motivation and theoretical analysis.**
>
> R1:Our paper introduces a novel framework, SSCF (Spiking Self-Cross Feature Network),  which advances SNN-based few-shot learning through two key innovations: (1) a Self-Feature Extractor that captures intra-class temporal correlations, overcoming SNNs' spatiotemporal feature extraction limitations, and (2) a Cross-Feature Contrastive module employing 4D convolutions and joint attention to optimize support-query alignment. Unlike existing SNN-FSL works focusing on spike-time encoding, our framework uniquely integrates self-correlation analysis with cross-attention mechanisms, offering more comprehensive feature learning while maintaining SNNs' energy efficiency advantages.
>
> **Q2:About the last Energy consumption model comparison.**
>
> R2:Thank you for highlighting this important aspect. Because of the real condition limitation in the lab, we are sorry that we do not have the conditions to use TPU yet. The experiments in our paper are conducted on NVIDIA 4060ti, with a performance of 22 TFLOPS. That is why we employ the energy consumption estimation based on the value 1.732mJ, which is consistent with [1]Efficient Event-based Semantic Segmentation with Spike-driven Lightweight Transformer-based Networks,2024. Here, we further supplement the new energy consumption estimation by referring the recent energy as [2]Spiking-physformer: camera-based remote photoplethysmography with parallel spike-driven transformer,2025.
>
> SOPs= *fr* × T × FLOPs
>
> Power(ANN)=4.6pJ × FLOPs
>
> Power(SNN)=0.9pJ × SOPs
>
> | Method     | SOPs   | FLOPs  | Energy   |
> |------------|--------|--------|----------|
> | ReNet(ANN) | -      | 4.84G  | 22.264mJ |
> | SSCF(ours) | 1.39G  | 0.13G  | 1.849mJ  |
>
> **Q3:Insufficient evidence for the energy efficiency experiment, CUB dataset not achieved SOTA.**
>
> R3:Thanks for your suggestion. The significant energy reduction stems from SNNs’ event-driven, sparse processing, aligning with neuromorphic principles. While performance is slightly lower than SOTA (-3.22% in 5-way-1-shot), our method bridges SNNs and ANNs, offering a low-power alternative with minimal accuracy loss—a key contribution for resource-constrained applications.Since SNNs have great spatiotemporal dynamics thanks to the spike-based neuron computation model, thus have advantages on processing event-based neuromorphic dataset, our experiments verified that the proposed SNNs’ few shot learning method remains and even improves these advantages. On event datasets (e.g., N-Omniglot), our model achieves 84.4% (20-way-1-shot), improving +20.3% over [3], leveraging SNNs’ spatiotemporal dynamics for event-data advantages. Moreover,on *mini*ImageNet, it also outperforms [4] by +7.3%, demonstrating competitive performance.Therefore, we emphasize that our model achieves competitive performance among these datasets.
>
>
> **Q4:Comparison with methods from recent years on the N-Omniglot dataset.**
>
> R4:First of all, thank you very much for the two references you provided. [5] Knowledge Transfer for SNNs (AAAI 2024) explores static-to-event domain adaptation, which is orthogonal to our work. Their experiments are conducted on event-based datasets, and our performance on N-Omniglot surpasses it. [6] Brain-Inspired Meta-Learning (TNNLS 2024) focuses on bearing fault diagnosis, due to the data employed in this paper focus on bearing part failure problem and is not public, which makes it difficult for me to compare it under the same experimental setting. Instead, we add discussion about this study in the related work section.
>
> | Dataset    | method              | backbone | Task         | acc  |
> |------------|---------------------|----------|--------------|------|
> | **N-Omniglot** | plain           | SCNN     | 20-way 1-shot | 63.4 |
> |            | Knowledge-Transfer[2] | SCNN     |              | 64.1 |
> |            | SSCF(ours)          | SCNN     |              | 67.8 |
> |            | SSCF(ours)          | VGGSNN   |              | 84.4 |
>
> I have carefully revised and proofread the spelling errors and citation errors. Thank you again for your valuable suggestions.
>
> **Reference**
>
> [1] Efficient Event-based Semantic Segmentation with Spike-driven Lightweight Transformer-based Networks, 2024.\
> [2] Spiking-physformer: camera-based remote photoplethysmography with parallel spike-driven transformer. Neural Networks, 2025\
> [3] An Efficient Knowledge Transfer Strategy for Spiking Neural Networks from Static to Event Domain, AAAI 2024.\
> [4] A two-stage spiking meta-learning method for few-shot classification[J]. Knowledge-Based Systems, 2024.\
> [5] An Efficient Knowledge Transfer Strategy for Spiking Neural Networks from Static to Event Domain, AAAI 2024.\
> [6] Brain-Inspired Meta-Learning for Few-Shot Bearing Fault Diagnosis, TNNLS 2024

---

> > ### Comment · Reviewer_c29L · 2025-04-09
> >
> > Thank you for the responses, which have addressed most of my concerns. I have re-rated the paper to “weak accept”.

---

### Official Review · Reviewer_VAei · 2025-03-14

**Overall Recommendation:** 2

**Summary:**

This paper focuses on leveraging spiking neural networks (SNNs) for few-shot learning (FSL) to enhance generalization ability and energy efficiency. The proposed method combines a self-feature extractor module and a cross-feature contrastive module to refine feature representation and reduce power consumption. Experimental results show that the proposed method improves the classification performance with low power consumption.

**Claims And Evidence:**

The majority of the claims presented in the submission are backed by clear evidence, though certain aspects are problematic:
(1) Generalization ability
The paper claims that the model shows strong generalization capabilities; however, the paper only evaluates it on three datasets, leaving it unclear how effectively the model would generalize to other, more diverse datasets, such as tieredimagenet, cifar, metadataset
(2) The relevance to FSL
The proposed method does not address the challenges faced in few-shot learning, and its relevance to few-shot learning tasks is limited. It seems this is a general work not specified for FSL.

**Essential References Not Discussed:**

No.

**Experimental Designs Or Analyses:**

(1) Dataset
The selected datasets are commonly used in few-shot learning, but with only three datasets, it is insufficient to validate the generalization capability and robustness of the proposed method.
(2) Performance Evaluation
Although comparisons are made with some SNN-FSL and SOTA few-shot learning methods, the methods being compared are outdated, rendering the comparison meaningless. There is no comparison with the latest and highest-performing methods. Even when compared, the proposed method does not surpass SOTA.
(3) Ablation Studies
The ablation experiments validate the effectiveness of the two modules, but only two datasets are tested, with the third dataset remaining unverified.

**Methods And Evaluation Criteria:**

The benchmark is designed for few-shot learning, but the proposed method has little relevance to few-shot learning.

**Other Comments Or Suggestions:**

(1) Please provide additional analyses and experiments to support your claim.
(2) Please correct the spelling and formatting errors.

**Other Strengths And Weaknesses:**

Weaknesses:
(1) The method proposed in the paper is not novel; it simply adds two simple attention-like modules to SNN.
(2) The proposed method does not address the challenges faced by existing SNN-FSL approaches, as mentioned in the paper, such as the struggle to effectively capture features from the complex spatiotemporal characteristics of inputs and the lack of cross-class feature comparison and classification.
(3) The proposed method does not address the issue of data scarcity faced in few-shot learning.
(4) The methods being compared are too outdated. It is necessary to include comparisons with the latest SOTA methods.
(5) Since the paper mentions that the proposed method improves generalization, yet the experiments lack evidence to substantiate this claim.
(6) The method diagrams are extremely blurry and contain significant amounts of blank space.
(7) There are spelling errors and blank citations.(Line 018, Line 121).

**Questions For Authors:**

Please refer to the comments above.

**Relation To Broader Scientific Literature:**

(1) Relevance to Few-Shot Learning
This paper aims to address the challenges faced in few-shot learning and serves as a supplement to few-shot learning research. However, the paper lacks an analysis of why the proposed method can tackle the challenges of few-shot learning, and its connection to few-shot learning is not clearly demonstrated.
(2) Relevance to Spiking Neural Networks
This paper highlights that SNNs exhibit potential in few-shot learning due to their event-driven nature and low energy consumption. Existing research, such as Efficient Online Few-Shot Learning and the SOEL system, has utilized SNNs for few-shot learning, addressing some issues but still facing challenges like capturing features effectively and performing cross-class comparisons. Building on these studies, this paper proposes an SNN framework that integrates a self-feature extractor module and a cross-feature comparison module to improve feature representation and reduce energy consumption, thereby enhancing the performance of SNNs in few-shot learning. This work serves as a supplement to existing SNN-based few-shot learning research. However, the proposed method does not resolve the issues mentioned in the paper, such as lack of cross-class feature comparison and classification.

**Theoretical Claims:**

The theoretical claims about the model and losses are correct. However, the proposed method is not novel; it merely employs two simple modules on SNN. While these modules enhance performance, there is, firstly, no reasonable theoretical analysis explaining why they are effective for few-shot tasks. Secondly, these modules bear a strong resemblance to self-attention and cross-attention mechanisms, although applied within a convolutional network framework.

---

> ### Author Rebuttal · Authors · 2025-03-30
>
> We sincerely appreciate the insightful feedback, which has helped us clarify our contributions and identify areas for improvement.
>
> **Q1:How to tackles the challenges of SNN’s few shot learning? Simply adds two simple attention-like modules ?**
>
> R1:We appreciate this critique and clarify that our key innovations go beyond simple "attention-like" modules: Our innovations transcend conventional attention mechanisms: The Self-feature extractor enhances intra-class representation through relational feature integration ("observe yourself for what" ); The Cross Feature Contrastive module dynamically models support-query relationships via joint attention maps. Together, they synergistically improve SNN few-shot learning through dual optimization of self-representation and cross-class comparison, not mere attention replication.
>
> **Q2: Cross-class comparison?**
>
> R2:Our method enables cross-class comparison via two specialized modules: A bottleneck-structured self-feature extractor capturing intra-class patterns while boosting inter-class discrimination; A 4D-convolution based cross-feature comparator generating attention maps for explicit query-support relationship modeling. This SNN-compatible design combines event-driven LIF dynamics with advanced feature extraction, achieving 98.9% accuracy (5-way 5-shot on N-Omniglot) with clear feature separation (t-SNE verified).
>
> **Q3:Data scarcity**
>
> R3:Data scarcity drives few-shot learning research, motivating our development of SNN-based solutions that leverage meta-learning and metric learning to extract meaningful patterns from limited samples while maintaining generalization capability. Our systematic validation demonstrates significant improvements through: comprehensive benchmarking against state-of-the-art SNN few-shot methods (see R4 results), and synergistic operation of our novel self-feature extraction and cross-feature comparison modules, which collectively enhance few-shot learning performance in SNNs while preserving their energy-efficient characteristics.
>
> **Q4:Compared to other recent advanced SNNs’ few shot learning methods.**
>
> R4:Thanks for your suggestion.We have added some of the latest SNN few-shot learning methods for comparison, The performance comparison is show as follows:
> | Dataset    | method              | backbone | Task         | acc  |
> |------------|---------------------|----------|--------------|------|
> | **N-Omniglot** | plain[1]           | SCNN     | 20-way 1-shot | 63.4 |
> |            | Knowledge-Transfer[2] | SCNN     |              | 64.1 |
> |            | SSCF(ours)          | SCNN     |              | 67.8 |
> |            | SSCF(ours)          | VGGSNN   |              | 84.4 |
>
> | Dataset          | method      | backbone               | Task         | acc  |
> |------------------|-------------|------------------------|--------------|------|
> | *mini*ImageNet  | OWOML [3]   | SNN-ResNet-12          | 5-way 1-shot | 45.2 |
> |                  | CESM [4]    | SNN-WideResNet-28-10   |              | 51.8 |
> |                  | MESM [4]    | SNN-WideResNet-28-10   |              | 53.6 |
> |                  | SSCF(ours)  | VGGSNN                 |              | 60.9 |
>
> **Q5:The Generalization ability of the proposed model.**
>
> Thanks for your suggestion. Here we validate the performance of our model on N-Omniglot, CUB, and *mini*ImageNet datasets, which are common-used validation settings as [5-7] . These datasets are recognized benchmark datasets in the FSL field, covering neuromorphic data, fine-grained classification, and general object classification scenarios, respectively.Meanwhile, in order to further verify the generalization performance and robustness of the model, we supplemented the performance of the model under different datasets and different noise levels as  Table 1 in Reviewer 4's comments.
>
> Thanks for your feedback. We have re-uploaded high-resolution images in the revision, and have carefully revised and proofread spelling errors and citation errors. We are sorry that we can not upload the figure during the rebuttal phase, but we really add more method details in the method d diagram to make it clear. Thank you again for your valuable suggestions.
>
> **Reference**
>
> [1] N-omniglot, a large-scale neuromorphic dataset for spatio-temporal sparse few-shot learning, 2022\
> [2] An Efficient Knowledge Transfer Strategy for Spiking Neural Networks from Static to Event Domain, AAAI 2024\
> [3] Fast on-device adaptation for spiking neural networks via online-within-online meta-learning, IEEE 2021\
> [4] A two-stage spiking meta-learning method for few-shot classification[J]. Knowledge-Based Systems, 2024\
> [5] Spatial-aware Metric Network via Patch-wise Feature Alignment for Few-shot Learning[J]. IEEE 2025.\
> [6] Hard-Positive Prototypical Networks for Few-shot Classification, IEEE 2025.\
> [7] EMNet: A Novel Few-Shot Image Classification Model with Enhanced Self-Correlation Attention and Multi-Branch Joint Module,2025

---

### Decision · Program_Chairs · 2025-05-01

**Decision:**

Accept (poster)

**Comment:**

This is a timely contribution to an important challenge in AI (few shot learning) through a biologically plausible spiking neural network. Overall, the authors provided strong responses to the reviewer's comments and the overall balance from the reviewers is that this is a valuable contribution to the field.